# Learn to Estimate Genetic Mutation and Microsatellite Instability with Histopathology H&E Slides in Colon Carcinoma

**DOI:** 10.3390/cancers14174144

**Published:** 2022-08-27

**Authors:** Yimin Guo, Ting Lyu, Shuguang Liu, Wei Zhang, Youjian Zhou, Chao Zeng, Guangming Wu

**Affiliations:** 1The Eighth Affiliated Hospital, Sun Yat-sen University, Shenzhen 518000, China; 2Center for Spatial Information Science, The University of Tokyo, Kashiwa 277-8568, Japan

**Keywords:** deep convolutional network, H&E slice, gene mutation prediction, microsatellite instability, colon carcinoma

## Abstract

**Simple Summary:**

Colorectal cancer is one of the most common malignancies and the third leading cause of cancer-related mortality worldwide. Identifying KRAS, NRAS, and BRAF mutations and MSI status are closely related to the individualized therapeutic judgment and oncologic prognosis of CRC patients. In this study, we introduced a cascaded network framework with an average voting ensemble strategy to sequentially identify the tumor regions and predict gene mutations & MSI status from whole-slide H&E images. Experiments on a colorectal cancer dataset indicated that the proposed method can achieve high fidelity in both gene mutation prediction and MSI status estimation. In our testing set, the AUCs for KRAS, NRAS, BRAF, and MSI were ranged from 0.794 to 0.897. The results suggested that the deep convolutional networks have the potential to assist pathologists in prediction of gene mutation & MSI status in colorectal cancer.

**Abstract:**

Colorectal cancer is one of the most common malignancies and the third leading cause of cancer-related mortality worldwide. Identifying KRAS, NRAS, and BRAF mutations and estimating MSI status is closely related to the individualized therapeutic judgment and oncologic prognosis of CRC patients. In this study, we introduce a cascaded network framework with an average voting ensemble strategy to sequentially identify the tumor regions and predict gene mutations & MSI status from whole-slide H&E images. Experiments on a colorectal cancer dataset indicate that the proposed method can achieve higher fidelity in both gene mutation prediction and MSI status estimation. In the testing set, our method achieves 0.792, 0.886, 0.897, and 0.764 AUCs for KRAS, NRAS, BRAF, and MSI, respectively. The results suggest that the deep convolutional networks have the potential to provide diagnostic insight and clinical guidance directly from pathological H&E slides.

## 1. Introduction

Colorectal cancer (CRC) is one of the most common lower gastrointestinal malignancies and is currently the third leading cause of cancer-related mortality worldwide [1,2]. Despite the over survival rate of colorectal cancer has increased in recent years due to the improved treatment strategies [3], distant metastasis is still a significant cause of high morbidity and mortality for CRC patients [4]. So far, various predominant environmental risk factors for the development of CRC have been identified, including diet, obesity, lack of physical activity, and inflammatory bowel disease [5]. However, a module formed by the interaction of multiple genetic alterations determines individual differences and tumor progression in CRC patients.

In the past decades, a deep understanding of molecular profiles has been more significant for selecting appropriate therapies for metastatic CRC patients [6]. Numerous frequent genetic mutations have been identified as critical drivers responsible for comprehensive therapeutic judgment and oncologic prognosis [7]. Mutations of RAS (i.e., exon 2, 3, and 4 of KRAS, exon 2 and 3 of NRAS) are considered negative predictors for targeted therapy with anti-EGFR monoclonal antibodies (e.g., cetuximab and panitumumab) [8,9]. Mutation of BRAF V600E is a worse prognostic biomarker. Patients with BRAF V600E mutation will be less likely to respond to treatment with cetuximab and panitumumab unless combined with a BRAF inhibitor [10,11]. Moreover, the microsatellite instability (MSI) status of CRC patients is also an important marker closely related to the assessment of prognosis, the efficacy of chemotherapeutic and immunity therapy [12,13]. Therefore, all metastatic CRC patients are suggested to detect the KRAS, NRAS, and BRAF mutations and MSI status according to the National Comprehensive Cancer Network (NCCN) clinical practice guidelines in oncology (Colon Cancer, Version 2.2021) [14].

The general diagnosis procedure of molecular pathology includes Sanger sequencing, Next-Generation Sequencing (NGS), ARMS-PCR, and digital PCR, etc. [15]. In recent years, the accuracy and sensitivity of those methods have been significantly improved. However, molecular detection remains limited by various factors such as sample quality, mutated gene abundance, and laboratory conditions. Moreover, in a short period of time, high testing prices are also a heavy burden for most families.

With the development of big data and deep convolutional network, artificial intelligence (AI)-assisted pathological diagnosis has attracted more and more attention. In 2018, Coudray et al. trained a deep convolutional neural network on Whole-Side Images (WSIs) to predict the cancer subtype and gene mutations in lung cancer [16]. Later, MSI status estimation of CRC from H&E histology was reported [17,18]. Furthermore, Skrede et al. exhibited a promising result in the survival risk interpretation of tumor patients based on artificial intelligence [19]. These methods have significantly extended the application capability of deep convolutional networks. However, genetic mutation prediction from H&E slices in CRC, which has more clinical significance in precision diagnosis, is still very challenging. To fulfill this demand and further explore the potential of H&E slides, we propose a cascaded deep convolutional framework to simultaneously generate gene mutation predicting and MSI status estimation using WSIs in colorectal cancer. The proposed method consists of two tumor region classification models, gene mutation& MSI status estimation models, and an average voting ensemble strategy. The effectiveness of the proposed method is demonstrated by a CRC dataset collected from GDC Data Portal and Eighth Affiliated Hospital, Sun Yat-sen University (see Section 2.1). In qualitative and quantitative evaluation, the proposed method reveals promising accuracy in tumor classification (0.939–0.976 AUC), gene mutation prediction (0.792–0.897 AUC), and MSI status estimation (0.764 AUC).

The main contributions of this study can be summarized as follows:We proposed a cascaded deep convolutional framework to simultaneously generate gene mutation prediction and MSI status estimation in colorectal cancer.We introduced a simple yet efficient average voting ensemble strategy to produce high fidelity gene mutation prediction and MSI status estimation of the WSI.We further analyzed the effectiveness of the number of features selected for model ensembling to understand its effects on the performances of deep CNN models.

The rest of the paper is organized as follows: Firstly, we present the datasets and methods used for this research in Section 2. Then, we illustrate the quantitative and qualitative results in Section 3. Finally, discussion and conclusion are presented in the Section 4 and Section 5, respectively.

## 2. Materials and Methods

### 2.1. Data

To explore the possibility of estimating somatic mutations and microsatellite instability (MSI) using Hematoxylin-Eosin(H&E) stained whole-slide image (WSI), we downloaded diagnostic slides and corresponding clinical data of the TCGA-COAD cohort from GDC Data Portal (https://portal.gdc.cancer.gov/projects/TCGA-COAD, accessed at 20 February 2022). The pre-compiled somatic mutation data and MSI status data were acquired from UCSC Xena (https://xenabrowser.net/datapages/, accessed at 10 March 2022) and MSIsensor-pro [20], respectively. The original WSIs were formated in a magnification ratio of either 20× or 40×. Prior to performing our experiments, we manually resize the 40× images to 20× using libvips (https://github.com/libvips/libvips) (see Figure 1A–D). There were 292 WSIs with corresponding somatic mutations and MSI statuses in the TCGA-COAD dataset. To achieve better generalization, we also collected the SYSU8H dataset with the cooperation of The Eighth Affiliated Hospital, Sun Yat-sen University. The selected pathological specimens were fixed in formalin, embedded in paraffin wax block, and cut by several consecutive slices in 3–5 um by a Leica HistoCore Autocut. Later, the slices were used for Hematoxylin-Eosin (H&E) staining, IHC staining, or gene sequencing, separately. Compared with the scanned H&E slices, the tumor areas for the sequencing slices are in micron-level drifts that tumor genomic heterogeneity among these slices is negligible. There were total 104 WSIs captured with 20× magnification ratio by PANNORAMIC 1000, 3DHISTECH Ltd.(see Figure 1E–H). Unlike next-generation sequencing (NGS) of TCGA-COAD, in the SYSU8H dataset, the genetic information was obtained by sanger sequencing. The binary masks of tumor areas of the WSIs were carefully annotated by experienced pathologists using QGIS (v3.22.7 LTR, https://qgis.org/).

As shown in Table 1, the 396 WSIs samples were randomly divided into training, validation, and testing groups with the ratios of 70%(278), 15%(59), and 15%(59), respectively. At 5× magnification WSIs, there were 283,126, 49,988, and 55,787 tiles within the corresponding training, validating, and testing set. At 10× magnification WSIs, 1,152,481, 203,183, and 2,275,595 tiles were within the corresponding training, validating, and testing set. In our experiment, the size of each tile was set to 512 × 512 pixels.

### 2.2. Methodology

In this study, we proposed a cascaded network framework to directly estimate somatic gene mutation and microsatellite instability status from the H&E stained whole-side image.

As shown in Figure 2, at the training stage, WSIs and corresponding binary masks of the training and validation set were partitioned into 5× or 10× tiles for training and validating the tumor classifier. The annotated tumor tiles and their somatic gene mutations or microsatellite instability (MSI) were used for training a binary classifier to discriminate wild type (i.e., W.T.) vs. mutant type (i.e., M.T.) of the gene or MSI-H vs. MSS/MSI-L, respectively. The top N highest probabilities of all tiles within a WSI were used to generate the final prediction for the patient.

Through several cycles of training and validation, the hyperparameters, including batch size, the number of iterations, and learning rate, were optimized with the Adam stochastic optimizer [21]. Subsequently, the predictions generated by the optimized models were evaluated using the WSIs of the test set (see details in Table 1). For performance evaluations, we carefully measured the area under the receiver operator characteristic (ROC) curve [22] and its confidence interval (CI) [23].

#### 2.2.1. Data Preprocessing

At first, the 396 pairs of whole-side images (WSIs) and their corresponding clinical records were shuffled and partitioned into three groups: training (70%), validating (15%), and testing (15%). Within each pair, a binary tumor mask of WSI was generated through polygon rasterization of its manually created tumor annotation. Later, a square window of 512 × 512 pixels was applied to the whole-side image and the corresponding tumor mask to extract paired tiles of WSI and mask. Then, each tile of WSI was labeled according to the positive ratio of pixels of the tumor mask. To focus on the tumor regions, tiles with positive ratios less than 80% were marked as 0. Otherwise, tiles were marked as 1. There were 388,901 and 1,583,259 tiles extracted from 5× and 10× magnification. As shown in Table 1, at 5× magnification, there were 283,126, 49,988, and 55,787 tiles within the training, validating, and testing set. While at 10× magnification, the number of tiles used for training, validation, and testing was 1,152,481, 203,183, and 2,275,595, respectively.

#### 2.2.2. Network Architectures

For simplicity and efficiency, we adopted an advanced convolutional neural network (CNN) architecture, i.e., EfficientNet [24], as a backbone for tumor classification and gene&MSI classification.

In 1998, Lecun et al. introduced the classic CNN architecture, LetNet-5 [25], which consists of two sets of convolutional & pooling layers, a flattening convolutional layer, and two fully-connected layers. The CNN reveals two important concepts, sparse connectivity and shared weights, significantly reducing memory occupation and promoting computational efficiency. With the growing complexity of the dataset and rapid development of computational capacity, computer scientists have proposed more advanced CNN architectures for better generalization capacity and computational efficiency [26]. These architectures significantly promote CNN performance by introducing well-designed novel strategies, such as network in network (i.e., NIN) [27], residual learning (i.e., ResNet) [28], inception architecture [29], and dense connection (i.e., DenseNet) [30]. Differ from the above-mentioned models, which mainly focus on model accuracy, the EfficientNet architecture is designed to get a present accuracy level with limited computational operations. The EfficientNet introduces a uniformed scaling method that scales all dimensions of depth, width, and resolution with a set of fixed scaling coefficients [24].

In our experiments, we chose an ImageNet-1K [31] pretrained EfficientNet B0 (https://pytorch.org/vision/master/models/generated/torchvision.models.efficientnet_b0.html, accessed at 4 March 2022) as the backbone for both tumor classification and Gene&MSI classification. As shown in Table 2, we introduced a dropout layer (*p* = 0.5) [32] to prevent overfitting. Then, we replaced the dimensions of fully-connected (FC) layer from 1280 × 1000 to 1280 × 1.

Subsequently, the activation function was changed from softmax to sigmoid.
(1)$$\begin{array}{cc} z_{i} & =b+\sum_{j=1}^{c}w_{j}\times x_{i,j}\\ p_{i} & =\frac{1}{1+e^{-z_{i}}} \end{array}$$

The *w* ∈ R${}^{c}$ and *b*∈ R${}^{1}$ denote the weights and bias, respectively. The range of prediction $p_{i}$ is limited to [0, 1].

Instead of binary cross entropy [33], we adopted focal loss [34] as our object function to focus learning on hard misclassified examples and address class imbalance. The equation can be formulated as:(2)$$\begin{array}{cc} p_{t} & =\left\{\begin{array}{cc} p_{i}, & \mathrm{if}\;y_{i}=1\\ 1-p_{i}, & \mathrm{if}\;y_{i}=0 \end{array}\right.\\ Loss_{focal} & =-{\left(1-p_{t}\right)}^{\gamma }log\left(p_{t}\right) \end{array}$$
where $p_{i}$ and $y_{i}$ is the *i*th prediction and corresponding ground truth. The value of $p_{t}$ is $p_{i}$ if the observation is in class 1; otherwise, the value is $1-p_{i}$. The γ (≥ 0) is a tunable focusing parameter which reduces the relative loss for well-classified examples (i.e., $p_{t}$ > 0.5) and puts more focus on hard, misclassified examples.

With all of the above layers being trained by mini-batch stochastic gradient descent (SGD) [35] to minimize the focal loss, the model learns how to map from the input 512 × 512 RGB image to a binary prediction.

#### 2.2.3. Model Ensemble

To make a decisive conclusion on the whole-slide-image (WSIs) using the separated predictions of 5× and 10× tiles, we introduced a simple yet efficient average voting strategy using the top N number of features to ensemble models. To ensure the high fidelity of selected features, a high threshold (i.e., 0.8) was used to filter out tiles with a low probability of being a tumor region. Later, tiles with a high probability of being tumor regions were passed to corresponding gene&MSI classification models to generate predictions of 5× tiles ($P_{x5}$) and 10× tiles ($P_{x10}$). Then, the top N highest probabilities of predictions from both 5× and 10× tiles were selected for the final estimation of the WSI ($P_{wsi}$). Finally, the $P_{wsi}$ and corresponding ground truth ($Y_{wsi}$) were used to calculate the area under the curve (AUC) for performance estimation.
(3)$$\begin{array}{cc} P_{topN} & =max(\left[P_{x5},P_{x10}\right],N)\\ P_{wsi} & =\frac{1}{N}\sum_{i=1}^{N}P_{topN} \end{array}$$

## 3. Results

A total of 396 colorectal cancer (CRC) patients with various gene mutations and MSI status from the SYSU8H and TCGA-COAD datasets were recruited in this study. The collected WSIs were randomly split into three sets: training, validation, and testing with the ratio of 70%, 15%, and 15%, respectively. The tiles extracted from the training and validation set wereused for training and optimizing hyperparameters of the proposed classification models. In order to estimate the performance of the proposed classification models, we have conducted heavy quantitative and qualitative comparisons on the testing set. All experiments were performed on the same dataset and processing platform.

### 3.1. Tumor Classification

The tumor regions annotated by the pathologist and probability maps generated by the tumor classification models using 5× and 10× tiles of WSIs are presented in Figure 3. Both 5× and 10× models display high fidelity in tumor recognition compared to manual annotations. Compared with the 5× model, the model trained with 10× tiles shows fewer false positives (e.g., orange and red patches outside the blue dashed curve of A, B, and C), fewer false negatives (e.g., blue and green patches inside the blue dashed curve of E and F), and better boundaries (e.g., around the blue dashed curve of A, B, and D). We selected 5 tiles from each of the four randomly selected whole slide images in the testing set, which present the highest probabilities to be the tumor regions according to our trained 5× or 10× tumor classification models (Figure 4 and Figure 5). The selected tiles show high consensus with the annotations by the pathologist. The receiver operator characteristic (ROC) curve and area under the curve (AUC), are used to evaluate the performance of tumor classification models using 5× and 10× tiles of the WSIs (Figure 6). The AUCs of 5× classification model have achieved 0.939 (95% CI of 0.937–0.940), 0.910 (95% CI of 0.905–0.914), and 0.959 (95% CI of 0.957–0.961) for training, validating, and testing set, respectively. Slightly better than the 5× model, the AUCs of 10× classification model are up to 0.971 (95% CI of 0.971–0.972), 0.973 (95% CI of 0.972–0.973), and 0.976 (95% CI of 0.975–0.977) for training, validating, and testing set, respectively. These values are consistent with our observation in Figure 3, Figure 4 and Figure 5.

### 3.2. Gene&MSI Classification

After model ensembling, the proposed method generates probabilities of gene mutations (i.e., KRAS, NRAS, and BRAF) and MSI status of every WSI.

As shown in Figure 7a–c, in the testing set, the proposed method reaches 0.792 (95% CI of 0.669–0.914), 0.886 (95% CI of 0.688–1.00), and 0.897 (95% CI of 0.800–0.994) AUCs for gene mutation predictions of KRAS, NRAS, and BRAF, respectively. In Figure 7d, our method shows high accuracy (i.e., 0.764 AUC, 95% CI 0.563–0.965) on the MSI status estimating in colorectal cancer.

To investigate the effect of the selected number of features (i.e., topN) used for model ensembling, we conducted a comparison experiment on the testing set using sequential values (i.e., [1, 3, 5, 7, 9]) of topN. Figure 8 shows the trend of the AUC values under sequential values of topN in the testing set. Among all values, the proposed method achieves the highest KRAS, NRAS, and BRAF gene mutation prediction accuracy while topN equals 7. In gene mutation predictions, as the value of topN increases, the AUC value will firstly increase and then decrease. In MSI status estimation, the AUC increases gradually as the value of topN increases. As the value of topN passes 7, the increment of AUC narrows down.

Figure 9 shows the top weighted tiles of whole slide images (WSIs) in gene mutation prediction and MSI status estimation by the proposed models.

## 4. Discussion

### 4.1. Regarding the Cascaded Framework

In recent years, deep convolutional networks have demonstrated their potential in computer-aided cancer identification using clinical images such as CT scan [36], ultrasonic [37], and MRI images [38]. Other than tumor recognization, a growing number of researches are trying to look deeper into microsatellite instability estimation [39,40], gene mutation prediction [41] or survival risk evaluation [19], which are vital for precision pathological diagnosis and treatment.

To the best of our knowledge, the proposed cascaded framework is the first end-to-end method that simultaneously generates gene mutation prediction and MSI status estimation using the whole slide image (WSI) in colorectal cancer. Our method can produce high-fidelity gene mutation prediction and MSI status estimation for each WSI through a simple yet efficient average voting strategy to ensemble models. Predicting the gene mutations (KRAS, NRAS, and BRAF) and MSI status from deep convolutional networks provides pathologists with a more convenient way to evaluate prognosis and guide medication. For example, advanced metastatic CRC patients with KRAS and NRAS mutations are not recommended to choose anti-EGFR monoclonal drugs (cetuximab and panimab) for treatment. The evaluation of BRAF mutation can stratify the prognosis and guide clinical treatment. Patients with BRAF genetic mutation are unlikely to respond to the treatment of cetuximab or panimab. MSI is a predictor of the efficacy of immune checkpoint inhibitors, CRC patients with MSI-H are more likely to benefit from the treatment of immune checkpoint inhibitors (e.g., pabolizumab). Qualitative and quantitative results of the experiment data demonstrated the effectiveness of our proposed framework. These results suggest that the deep learning models have the potential to provide diagnostic insight and clinical guidance directly from pathological H&E slides. Additionally, as the gene mutation prediction and MSI status estimation are directly computed from histopathology H&E slides, in principle, the proposed method should apply not only to colorectal cancer but also to other malignant cancers (e.g., lung, breast, and liver cancer).

### 4.2. Accuracies, Uncertainties, and Limitations

The proposed framework revealed high values of area under the curve (AUC) in both tumor classification and gene&MSI classification tasks. In tumor classification, the 5× and 10× classification models achieved 0.959 (95% CI of 0.957–0.961) and 0.976 (95% CI of 0.975–0.977) AUCs in the testing set, respectively. The values show a very close judgment between the pathologist and the proposed method, which suggest that the AI-algorithm can potentially serve as a pre-screening tool. The performance will be further evaluated using a larger dataset with multiple tissue samples collected from varied pathology departments.

In gene mutation prediction, the proposed method achieved 0.792 (95% CI of 0.669–0.914), 0.886 (95% CI of 0.688–1.00), and 0.897 (95% CI of 0.800–0.994) AUCs for gene mutation predictions of KRAS, NRAS, and BRAF, respectively. Because of the extremely biased ratio of mutant type / wild type distribution (i.e., 15 vs. 381 of NRAS, 43 vs. 353 of BRAF), the value of AUCs fluctuates in a large range within 95% confidence interval (see details in Figure 7). In terms of MSI status estimation, recent researches [39,40] had reported higher performance than ours(i.e., 0.764 AUC, 95% CI 0.563–0.965). Compared with these methods, our method is able to simultaneously generate gene mutation prediction (KRAS, NRAS, and BRAF) and MSI status estimation, which are all mandatory for metastatic CRC patients. As for future clinical application, improving the accuracy level of our algorithm remains one of the main future goals.

With the current cascaded classification-based scheme, the models are trained to generate tile-to-label predictions using features extracted from sequential convolutional layers. The lack of internal connectivity with adjacent tiles within the same WSI might lead to partial misclassification (e.g., red patches outside the blue dashed curve and green patches within the blue dashed curve in Figure 1B,D). Since the models are trained and optimized separately, the proposed framework requires extra computational time and storage for training and saving checkpoints of multiple models. Considering the computational efficiency, a unified model with shared parameters and object functions should be explored in further work.

Considering the type of H&E used for staining, varied types of hematoxylin have certain differences in stability, durability, and dyeing time, which may lead to distinct visual patterns. In the SYSU8H dataset, the H&E slices were stained using an identical form of hematoxylin (i.e., Harris hematoxylin) to make sure both the nucleus and cytoplasm can be clearly visible and discriminated. Due to the fact that the TCGA-COAD dataset was collected from multiple centers, the forms of hematoxylin used for staining were very likely to be different. However, as shown in Figure 3, in tumor classification, prediction accuracies among slices were not so significant. The result indicates that our method can be adapted to different forms of H&E staining approaches.

Another issue that should not be ignored is the tumor heterogenity of the primary and metastatic lesions. Clinically, whether it is pathological diagnosis or target gene detection, the tumor specimen of the primary lesion is the first choice. However, there may be discrepancies in the gene mutation between the primary and metastatic tumor. For advanced metastatic tumors, when the target gene mutation of the primary tumor is negative, the target gene detection of the metastatic tumor can be carried out if conditions permitted, which can increase the opportunity for patients to receive one more targeted drug treatment. In this study, limited by the publicly available clinical samples attached with the gene mutation information of the primary tumor and the corresponding metastases, our method focused exclusively on primary tumors. Further evaluation is still necessary to clarify the reliability and generalization of our model performance.

## 5. Conclusions

For colon carcinoma, we design a cascaded deep convolutional framework to simultaneously generate gene mutation predicting and MSI status estimation based on the whole-slide images. The proposed method introduces a simple yet efficient average voting ensemble strategy to produce a high-fidelity prediction of the WSI. In gene mutation&MSI status classification task, the proposed method achieves 0.792 (95% CI of 0.669–0.914), 0.886 (95% CI of 0.688–1.00), 0.897 (95% CI of 0.800–0.994), and 0.764 (95% CI 0.563–0.965) AUCs for KRAS, NRAS, BRAF, and MSI, respectively. These results suggest that the deep learning models have the potential to provide diagnostic insight and clinical guidance directly from pathological H&E slides. We plan to improve the architecture of the framework and apply it to other data sources to achieve better generalization capacity and diagnostic reliability.

## Figures and Tables

**Figure 1 cancers-14-04144-f001:**
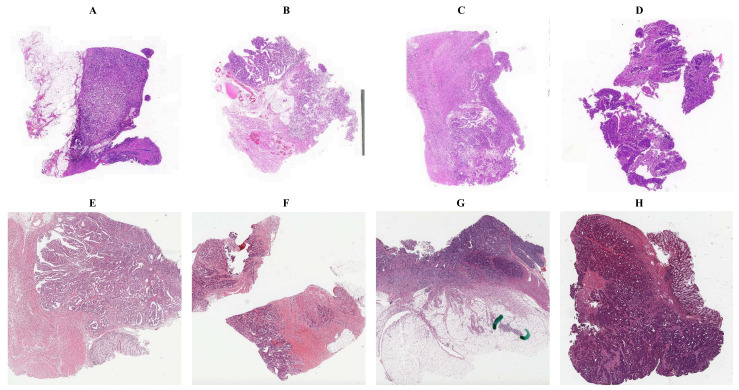
Representative H&E stained whole-side images (WSIs) from SYSU8H and TCGA-COAD dataset. The (**A**–**D**) and (**E**–**H**) samples are randomly selected from SYSU8H and TCGA-COAD datasets, respectively.

**Figure 2 cancers-14-04144-f002:**
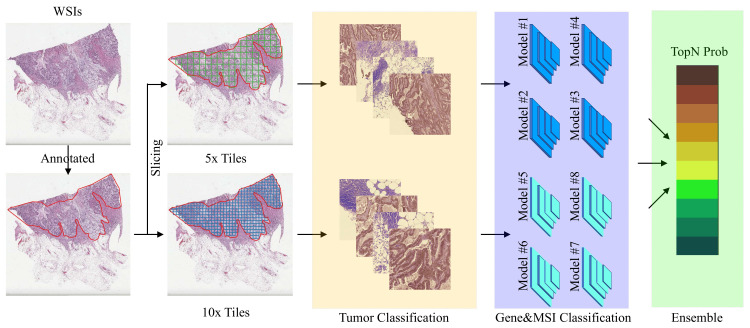
Experimental workflow for estimating somatic gene mutation and microsatellite instability with H&E stained whole-side images. The 5× or 10× tiles from WSIs will be accessed by a tumor classifier, a gene&MSI classifier, and a TopN ensemble classifier.

**Figure 3 cancers-14-04144-f003:**
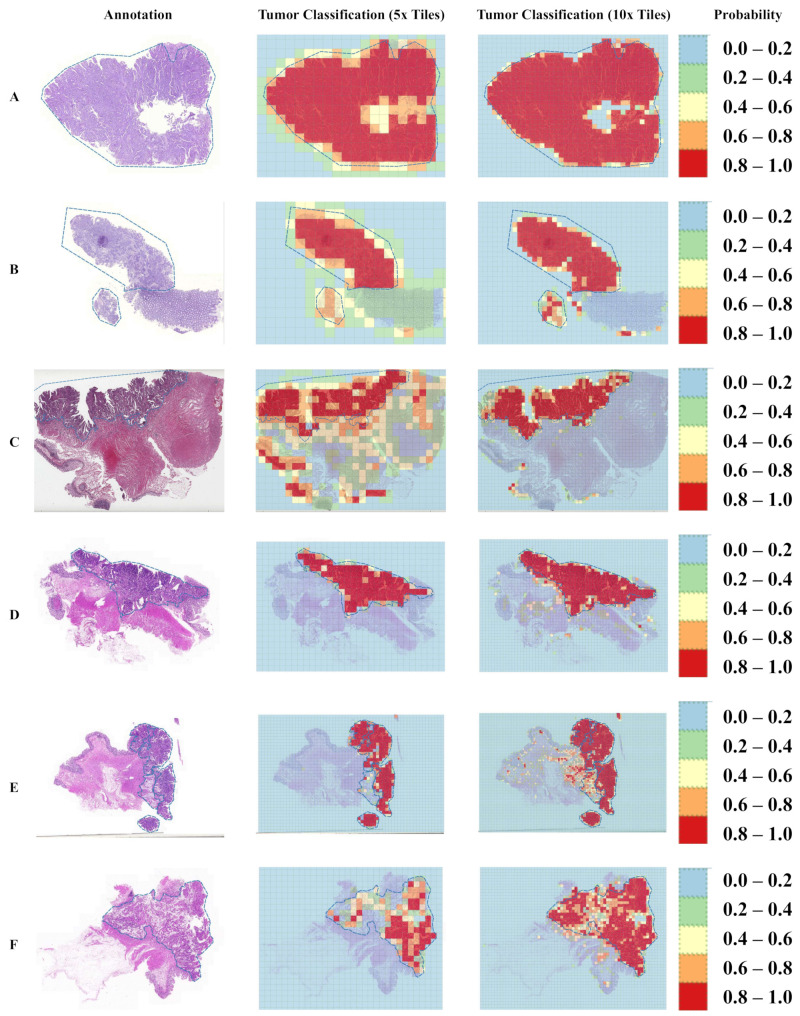
Probability maps of tumor classification using 5× and 10× tiles of the whole slide images (WSIs). The annotations created by the pathologist are marked with the blue dashed curve. The probability values are categorized into five groups with different color representations. The (**A**–**C**) and (**D**–**F**) samples are randomly selected from the testing set of TCGA-COAD and SYSU8H, respectively.

**Figure 4 cancers-14-04144-f004:**
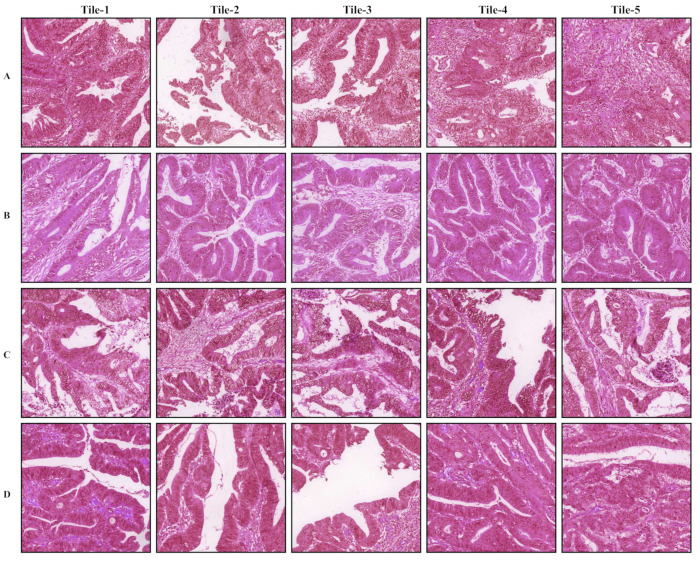
Representative tiles of tumor classification using 5× tiles of the whole slide images (WSIs). The (**A**–**D**) samples are randomly selected from the testing set. In each row, tiles 1–5 are patches from the same WSI.

**Figure 5 cancers-14-04144-f005:**
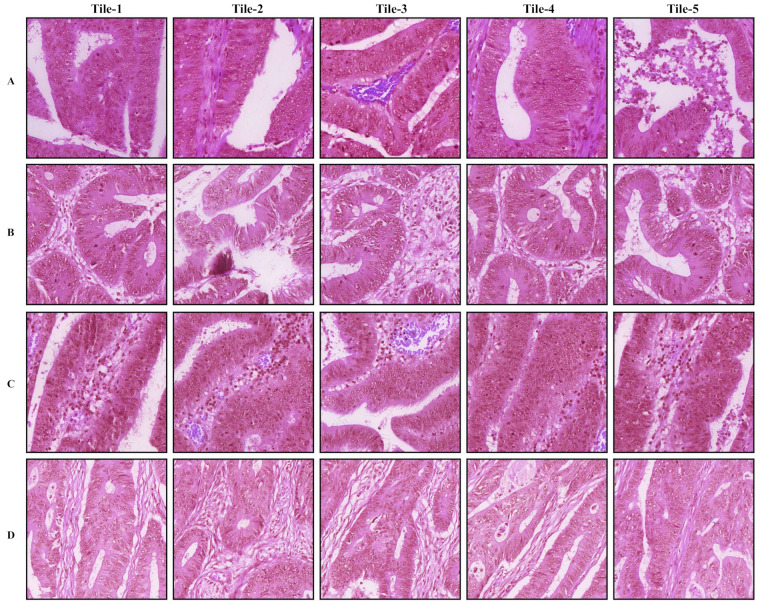
Representative tiles of tumor classification using 10× tiles of the whole slide images (WSIs). The (**A**–**D**) samples are randomly selected from the testing set. In each row, tiles 1–5 are patches from the same WSI.

**Figure 6 cancers-14-04144-f006:**
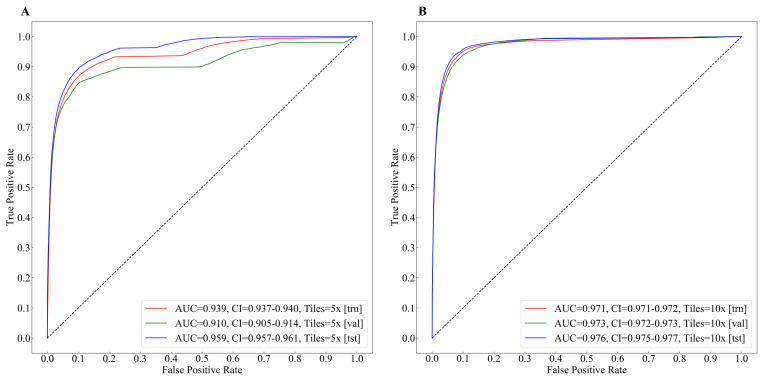
The receiver operator characteristic (ROC) curve and area under the curve(AUC) of tumor classification using 5× and 10× tiles of the whole slide images (WSIs). (**A**) The curves of training, validating, and testing set using 5× tiles. (**B**) The curves of training, validating, and testing set using 10× tiles.

**Figure 7 cancers-14-04144-f007:**
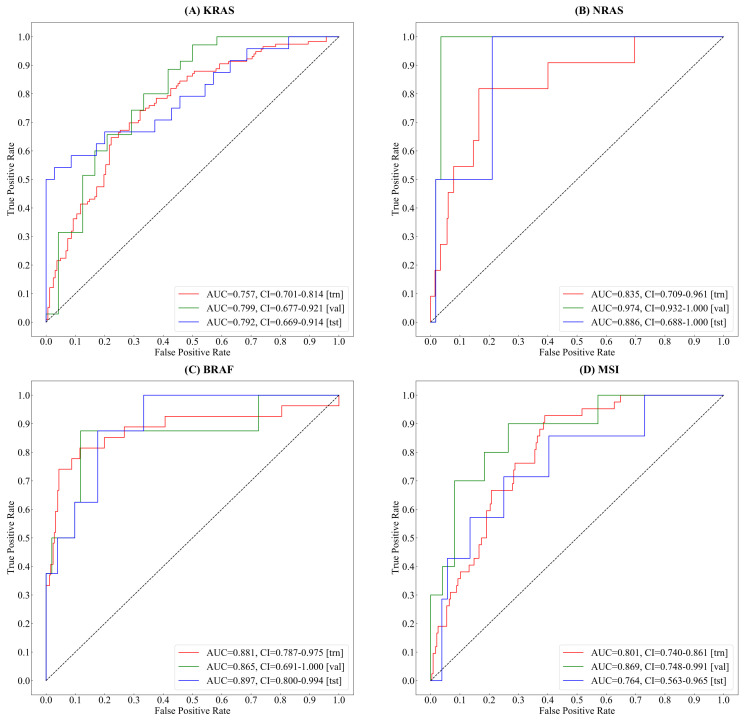
The receiver operator characteristic (ROC) curve and area under the curve(AUC) of Gene&MSI classification using 5&10× tiles of the whole slide images (WSIs). (**A**) The curves of KRAS gene mutation classification. (**B**) The curves of NRAS gene mutation classification. (**C**) The curves of BRAF gene mutation classification. (**D**) The curves of MSI status classification.

**Figure 8 cancers-14-04144-f008:**
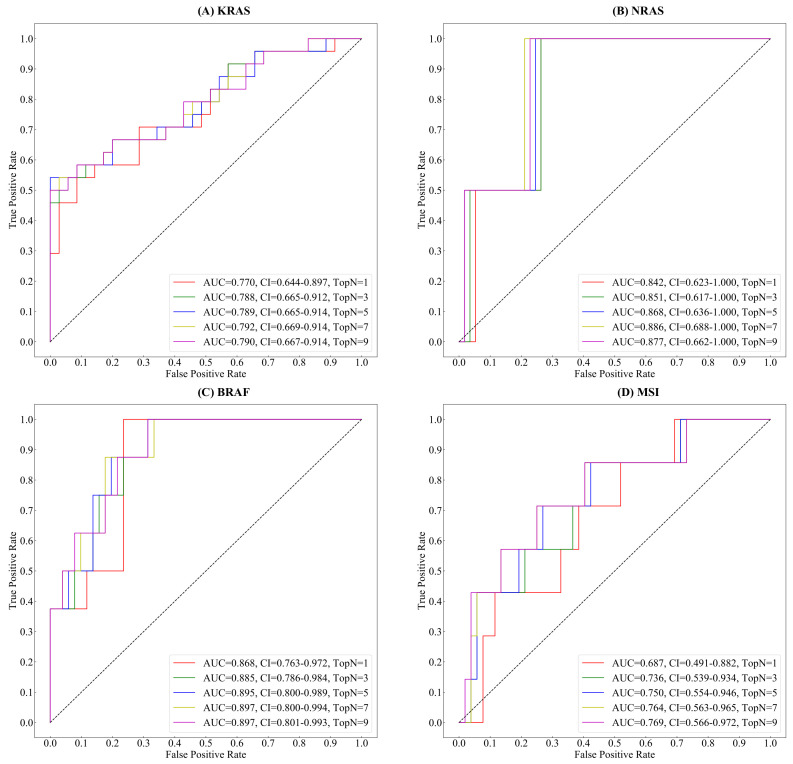
The receiver operator characteristic (ROC) curve and area under the curve(AUC) of Gene&MSI classification using sequential values of topN. (**A**) The trend of KRAS gene mutation classification. (**B**) The trend of NRAS gene mutation classification. (**C**) The trend of BRAF gene mutation classification. (**D**) The trend of MSI status classification.

**Figure 9 cancers-14-04144-f009:**
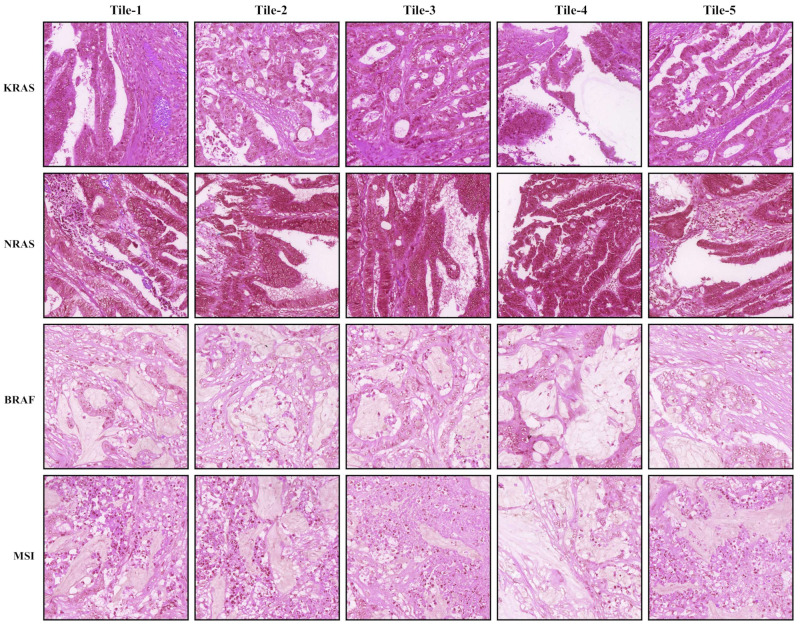
Top weighted tiles of whole slide images (WSIs) in gene mutation and MSI status estimation. In each row, tiles 1–5 are either 5× or 10× tiles extracted from the same WSI.

**Table 1 cancers-14-04144-t001:** Distribution of patients and whole-side images samples.

		WSIs			
		Train (n = 278)	Val (n = 59)	Test (n = 59)	Overall (n = 396)
Age(year)	Min.	22	29	36	22
	Max.	90	90	90	90
	Median	65.5	67	68	66
Gender	Male	138	30	33	201
	Female	140	29	26	195
KRAS	W.T.	162	24	35	221
	M.T	116	35	24	175
NRAS	W.T.	267	57	57	381
	M.T	11	2	2	15
BRAF	W.T.	251	51	51	353
	M.T.	27	8	8	43
MSI	MSI-H	236	49	52	337
	MSS/MSI-L	42	10	7	59
Tiles	5× Mag.	283,126	49,988	55,787	388,901
	10× Mag.	1,152,481	203,183	227,595	1,583,259

**Table 2 cancers-14-04144-t002:** The backbone network for both tumor classification and Gene&MSI classification. Each row describes the stage, operation, input resolution, output channel, and the number of layers.

Stage	Operator	Resolution	Channels	Layers
0		512 × 512	3	0
1	Conv3 × 3	512 × 512	32	1
2	MBConv1, k3 × 3	256 × 256	16	1
3	MBConv6, k3 × 3	256 × 256	24	2
4	MBConv6, k5 × 5	128 × 128	40	2
5	MBConv6, k3 × 3	64 × 64	80	3
6	MBConv6, k5 × 5	32 × 32	112	3
7	MBConv6, k5 × 5	32 × 32	192	4
8	MBConv6, k3 × 3	16 × 16	320	1
9	Conv1 × 1&Pooling	16 × 16	1280	1
10	Dropout&FC	1280 × 1	1	1

## Data Availability

Not applicable.

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
