# Peer review of "Learn to Estimate Genetic Mutation and Microsatellite Instability with Histopathology H&E Slides in Colon Carcinoma"

_cancers, 2022, doi:10.3390/cancers14174144_

Round 1

Reviewer 1 Report

Thank you for the opportunity to review this interesting paper.

The authors have performed a study on 396 patients with colorectal cancer to estimate genetic mutations and microsatellite instability on HE slides.

The results are very interesting and timely. 

The methods are well described and performed.

I have no relevant concerns for this study.

Reviewer 3 Report

The era of pathomics started several years ago and the new papers related to different localizations pathomics-based algorithms release every day. The authors proposed the deep learning based-approach for colorectal cancer key mutations (KRAS, NRAS, BRAF and MSI) status evaluation on whole slide imaging analysis stained with standard histological stain (h/e). Previously the investigations related to KRAS and MSI status prediction have already been published. Unfortunately the most recent publications were not cited in the article: for example, Artificial intelligence for detection of microsatellite instability in colorectal cancer-a multicentric analysis of a pre-screening tool for clinical application (ESMO Open. 2022 Apr;7(2):100400); Artificial Intelligence for Predicting Microsatellite Instability Based on Tumor Histomorphology: A Systematic Review. Int J Mol Sci. 2022 Feb 23;23(5):2462; Artificial Intelligence for Histology-Based Detection of Microsatellite Instability and Prediction of Response to Immunotherapy in Colorectal Cancer. Cancers (Basel). 2021 Jan 21;13(3):391). At the same time the authors have to compare their results with previously accepted ones. Actually, all these genes simultaneously were analyzed for the first time, although all of them have already used for the tumor molecular status identification with AI-based approaches.

It should be mentioned that KRAS, BRAF, NRAS and MSI mutations status determines potential survival rate and impacts the treatment strategy significantly. Thus, proposed approach has a dramatic clinical implementation. The accuracy of developed neural network is sophisticated: from 0,76 to 0,90. Even more prominent results accepted for tumor tissue verification: from 0,91 to 0,98. I suppose, that the authors should further test this algorithm with other tissue samples (from different pathology departments, cities and even countries) in future, because the accuracy higher then 0,9 can be compared with mutational analysis. So, the proposed algorithm could not even facilitate the stratification of the patients’ cohorts who require further mutation analysis (pretesting) but even replace it in the hospitals lacking the genetic an immunohistochemistry equipment. I recommend the authors to describe the accuracy and disadvantages of mutation analysis with standard methods to compare these features with proposed artificial intelligence-based algorithm and to show the potential benefits of the neural network more clearly.
